# Immunotherapy Plus Locoregional Therapy Leading to Curative-Intent Hepatectomy in HCC: Proof of Concept Producing Durable Survival Benefits Detectable with Liquid Biopsy

**DOI:** 10.3390/cancers15215220

**Published:** 2023-10-30

**Authors:** Roma Raj, Chase J. Wehrle, Nihal Aykun, Henry Stitzel, Wen Wee Ma, Smitha Krishnamurthi, Bassam Estfan, Suneel Kamath, David C. H. Kwon, Federico Aucejo

**Affiliations:** 1Cleveland Clinic Foundation, Department of Hepato-Pancreato-Biliary & Liver Transplant Surgery, Digestive Diseases and Surgery Institute, Cleveland, OH 44195, USA; roma.raj@utsouthwestern.edu (R.R.); nihal.aykun@duke.edu (N.A.); hjs85@case.edu (H.S.); kwonc2@ccf.org (D.C.H.K.); aucejof@ccf.org (F.A.); 2Cleveland Clinic Foundation, Department of Hematology and Oncology, Taussig Cancer Institute, Cleveland, OH 44195, USA; maw4@ccf.org (W.W.M.); krishns3@ccf.org (S.K.); estfanb@ccf.org (B.E.); kamaths@ccf.org (S.K.)

**Keywords:** advanced hepatocellular carcinoma, immunotherapy, locoregional therapy, biomarkers, alpha fetoprotein

## Abstract

**Simple Summary:**

Immunotherapy is emerging as an improved systemic treatment for select patients with advanced unresectable hepatocellular carcinoma. An objective response is reported in 30% of patients, yet a complete response allowing for curative-intent surgery is rare. Locoregional therapies seem to show synergistic effects with immunotherapy, though this effect has not been scientifically reported. We report a cohort of patients showing a complete response to this combination immunotherapy + LRT and aim to present this as a proposed treatment approach for locally unresectable disease. We also report how liquid biopsy using ctDNA was cleared using this approach and discuss how this testing modality may assist patients with this type of disease.

**Abstract:**

Background: Immunotherapy has emerged as an improved systemic treatment for select patients with advanced unresectable HCC. Objective response is reported in 30% of patients, yet complete response (pCR) allowing for curative-intent resection is rare. Locoregional therapies (LRTs) seem to show synergistic effects with immunotherapy, though this effect has not been scientifically reported. We report a cohort of patients showing pCR to immunotherapy + LRT as a proof of concept for the proposed treatment approach for locally unresectable HCC. Methods: Patients with unresectable HCC treated with immunotherapy as an intended destination therapy from 2016 to 2023 were included. The electronic health record was queried for oncologic information, locoregional therapies, surgical interventions, and long-term outcomes. Circulating tumor DNA (ctDNA) testing was obtained using Guardant360, and tumor mutational burden (TMB) was defined as the number of somatic mutations per megabase. Results: Ninety-six patients with advanced HCC received immunotherapy + LRT as a destination therapy. In total, 11 of 96 patients showed a complete response according to mRECIST criteria. Four of these (36.4%) ultimately underwent curative-intent resection. The median follow-up was 24.9 (IQR 15.6–38.3) months. Overall survival rates in those with complete response at 1, 3, and 5 years were 100%, 91%, and 81.8%, respectively, which were significantly improved compared to those of the cohort not achieving pCR (*p* < 0.001). All four patients undergoing immunotherapy + LRT followed by curative-intent hepatectomy have no evidence of disease (NED). Of those undergoing surgery, ctDNA was cleared in 75% (n = 3), providing an additional objective measurement of complete response. All four patients were TMB+ before beginning this treatment course, with three being TMB-, indicating stable and complete disease response. Conclusions: Immunotherapy + locoregional therapy can help downstage a significant proportion of patients with initially unresectable HCC, allowing for curative-intent surgery. The survival benefit associated with complete response seems durable up to 3 years after achieving this response. ctDNA measurement was converted from positive to negative in this cohort, providing additional indication of response.

## 1. Introduction

Hepatocellular carcinoma (HCC) is the most common primary liver tumor and one of the leading causes of cancer-related death worldwide [1]. Approximately 60% of HCC cases are diagnosed at an advanced stage and cannot be treated with surgery or locoregional therapies [2]. These patients received systemic therapy [3]. Immunotherapy has emerged as an improved systemic treatment for select patients with advanced HCC [4]. The current standard first-line therapy is a combination of the immune checkpoint inhibitor (ICI) atezolizumab (PDL1 inhibitor) and the vascular endothelial growth factor inhibitor (VEGF) bevacizumab [4,5,6]. This treatment offers an objective response in only about one-third of patients, and survival remains limited; however, a complete response is occasionally observed, allowing liver resection [4,5,6]. Additionally, some studies suggest that locoregional therapies show synergistic effects with immunotherapy in some cases and improve response rates and outcomes [7].

Circulating tumor DNA (ctDNA) is an emerging tool for HCC. In other cancers, ctDNA has been shown to predict response to systemic therapy and to predict recurrence after resection [8,9,10,11]. One key component of ctDNA is tumor mutational burden (TMB). TMB can be obtained from both tissue and blood; the tissue-based results have been associated with worse prognosis in HCC, and with predicting response to immunotherapy [10,12,13,14,15,16]. The liquid biopsy (LB)-based approach has not been validated in this cohort, and results of ctDNA have not yet been reported in conjunction with immunotherapy results.

Herein, we present the largest cohort of patients yet reported that showed exceptional response to systemic immunotherapy with or without concurrent locoregional interventions, demonstrating prolonged survival despite presenting with advanced-stage HCC.

## 2. Methods

The prospectively collected comprehensive liver database was queried for patients with advanced HCC who received immunotherapy after obtaining approval from our Institutional Review Board (IRB). Data collected from the electronic medical records included cross-sectional images, laboratory values, serum biomarkers, pathology reports, and follow-up and recurrence information. The aim of this study was to describe the outcomes in patients who responded to systemic treatment with immunotherapy. Patient response to immunotherapy was evaluated using the modified Response Evaluation Criteria in Solid Tumors (mRECIST) scoring system. Atezolizumab/bevacizumab is preferred as a first-line immunotherapy at our institution, unless it is unavailable due to cost or manufacturing issues. Nivolumab is also sometimes used in refractory cases that have already received atezolizumab/bevacizumab, or in cases where there is an issue with tolerance of the atezo/bev regimen. Changes in alpha-fetoprotein (AFP) levels were also considered. Patient survival was also recorded. IRB approval was obtained; consent was waived due to the retrospective nature of the review.

At the initiation of this study, ctDNA was assessed at the discretion of the treating physician without a clear institutional protocol. Because of this variable timing, contains a timeline annotation of the timing of ctDNA draws for each included patient with exceptional response. Our current institutional protocol for patients with HCC undergoing curative-intent surgery was developed and implemented in January 2021. Patients now receive standard-of-care ctDNA draws within 30 days preoperatively, 30–60 days postoperatively, and q3–6 months thereafter (Figure 1). ctDNA was obtained using the Guardant360 platform (Guardant Health, Redwood, CA, USA). The panel reports specific mutations for a pan-cancer 83 gene panel, and assesses mutations over 500 genes to establish a genomic footprint for the calculation of tumor mutational burden (TMB). Blood draws were performed in our facilities’ outpatient laboratory or coordinated by using mobile phlebotomy at patients’ homes based on patient preference. TMB in the included assay is defined as the number of somatic mutations per megabase. No detectable mutations are considered “TMB-ND” and >0 mutations are considered “TMB-detectable” Note that tissues of primary tumors were not sequenced in this study, though this has since become our standard practice.

Descriptive statistics were used to summarize patient demographics, clinical characteristics, and treatment details. Continuous variables were presented as means with standard deviations or medians with interquartile ranges, depending on their distribution. Categorical variables were presented as frequencies and percentages. Kaplan–Meier survival curves were constructed to estimate OS in patients with exceptional responses to immunotherapy. All statistical analyses were conducted using SPSS Version 29.0 with statistical significance defined as *p* < 0.05.

## 3. Results

Between 2016 and 2023, 96 patients with advanced HCC underwent systemic treatment with immunotherapeutic agents. In total, 11 of these 96 patients showed complete response according to the modified Response Evaluation Criteria in Solid Tumors (mRECIST). Nine (81.8%) of the eleven complete responders were male. The mean age at diagnosis was 66.7 ± 9.7 years (Table 1). Five patients (45.4%) had cirrhosis, most commonly due to HCV infection (n = 3, 27%). Three (27%) patients had metastatic disease (stage IVB), one (9%) had nodal disease (stage IVA), and four (36%) had major vascular invasion (stage IIIB) at the time of starting immunotherapy. The mean pre-treatment AFP was 28,035.25 ± 58,034.21 ng/mL, with the highest noted AFP being 200,000 ng/mL. Three patients had a history of surgical resection of their cancer and were treated with immunotherapy for recurrent disease. Four patients (36%) did not receive any locoregional or systemic therapy prior to or concurrent with immunotherapy (Table 2). The timeline of treatment, ctDNA testing, and disease response in the 11 patients is provided (Figure 1). A summary of previously presented case reports on this topic is also provided (Table 3).

The mean number of cycles received was 15 ± 12 (min = 1, max = 45). Three patients experienced immunotherapy-related adverse events, and one patient discontinued treatment. The mean post-treatment AFP was 32.42 ± 49.39 ng/mL, with the highest value being 158.6 ng/mL.

The median follow-up was 24.9 (IQR 15.6–38.3) months. In total, 9 of 11 patients were alive at the time of this report. One patient died due to bevacizumab-related complications, while the second patient experienced recurrence 15 months after stopping immunotherapy and died of metastatic disease. Four patients underwent resection after successful downstaging, while five patients remained unsuitable for surgery due to their comorbidities. One patient had a complete pathologic response (pCR) while three had <50% of viable tumors, corresponding to tumor response grade (TRG) 1. Overall survival (OS) rates at 1, 3, and 5 years were 100%, 91%, and 81.8%, respectively. 

Four patients underwent surgical resection of the remaining disease with available pre- and post-treatment ctDNA levels. All four patients had identifiable TMB on their pre-treatment ctDNA. However, three of four patients (75%) were rendered TMB-ND after their resection (Table 4). All three of these patients are currently alive with NED, with a mean follow-up of 43.2 months.

## 4. Discussion

To our knowledge, this represents the largest series of patients describing a significant to complete response to neoadjuvant immunotherapy for advanced-stage HCC. This proof-of-concept manuscript demonstrates a high rate (>10%) of patients who presented with unresectable HCC and were able to achieve prolonged survival after multimodal treatment, including systemic immunotherapy, ablation, transarterial chemoembolization (TACE), transarterial radioembolization (TARE) with Y-90, and stereotactic body radiation therapy (SBRT), and underwent surgery after successful downstaging. We also show that systemic immunotherapy + LRT downstaging can allow for eventual curative-intent resection even in very advanced cases. Finally, we show that this sequence leading to curative-intent resection can lead to clearance of previously identifiable blood-based TMB, further indicating successful systemic treatment.

Although surgery offers the best outcomes in patients with HCC, only 30–40% present with resectable disease at the time of diagnosis [36,37]. Furthermore, some patients cannot undergo surgery because of comorbidities and compromised liver function [38]. Immunotherapy has ushered in a new era in the treatment of advanced HCC in such patients, with an objective response achieved in approximately 30% of patients and a smaller subset of just 1–5% showing complete response [39,40]. In patients with good functional status and preserved liver function, resection of the residual tumor can be achieved. In patients who are not candidates for surgery because of compromised liver function and other comorbidities, systemic immunotherapy with or without the aid of LRTs can offer prolonged overall or progression-free survival [4,5,6]. Our study further demonstrates this utility of immunotherapy, with a salvage therapy rate of over 10%. Preliminary work has focused on clinical predictors of response to immunotherapy, but additional work should focus on identifying those that will experience the described robust response [41].

Some studies suggest that locoregional intra-arterial and ablative treatments (LRTs) provide a synergistic action with systemic immunotherapy and improve tumor response rates [42,43,44,45,46]. For example, Singh et al. have shown that LRT alone has immunomodulating effects that may sensitize the tumor to the effects of systemic immunotherapy [45]. This synergy was reflected in the results of our study, given the high rate of systemic response to immunotherapy and LRT. In our cohort, bevacizumab was held four weeks before LRT as per safety recommendations, and atezolizumab or nivolumab was held only if the scheduled LRT coincided with the day of the ICI cycle. Thus, these results also confirm that concurrent treatment with LRTs in patients receiving immunotherapy is feasible and can be administered without long breaks between doses. Sangro et al. report in a review and meta-analysis a complete response rate of 1–5% with immunotherapy for HCC [40]. There is also a known rate of recurrence after treatment with LRT alone, as demonstrated by Facciorusso et al. and others [47,48]. The addition of immunotherapy could theoretically help lessen the likelihood of post-LRT recurrence, as evidenced by the subset of patients who had LRT + immunotherapy alone and are currently NED, though this cannot be directly proven in this study. Given the small sample size, this study is intended to serve as a proof of concept. Thus, though our findings are not definitive, the >10% complete response rate in our study may provide additional evidence that combined LRT and immunotherapy has significant potential for downstaging to potential curative-intent resection in advanced HCC.

Immune checkpoint inhibitors also have a milder side effect profile than that of known tyrosine kinase inhibitors previously used in the management of advanced HCC [4,6]. Only one patient in our study discontinued treatment due to immunotherapy-related grade 4 hepatitis. However, it is interesting to note that this patient received only one cycle of atezolizumab and achieved complete response. One patient developed immune-related hepatitis, and one patient developed immune-related arthritis. These patients were managed by holding immunotherapy and prescribing steroids, respectively, and resumed treatment after the successful control of adverse effects. One patient receiving a combination of atezolizumab and bevacizumab died due to gastrointestinal hemorrhage. Bevacizumab is reported to increase the risk of gastrointestinal hemorrhage or perforation fourfold in patients with advanced disease [5,49]. Immunotherapy was generally well tolerated in this study, but these side effects are of important clinical significance for providers.

ctDNA-based liquid biopsy is a potentially useful tool for judging the response to treatment as well as identifying potential biomarkers in addition to TMB, which can be associated with response or resistance to immunotherapy [8,9,10,11]. New research has also investigated the utility of ctDNA as a method of detecting minimal residual disease (MRD), a testing approach that identifies the smallest number of remaining tumor cells or tumor DNA needed to create a disease recurrence. This concept has been discussed and preliminarily reported in HCC, but there no globally accepted approach for this at this time [50,51,52]. TMB is not specifically an MRD marker, but low or nondetectable TMB is associated with reduced progression [53,54], and thus, the conversion of TMB-detectable to nondetectable in our patients may provider further support for this approach.

This study has limitations. While this is the largest published series, we report a very small sample size of patients with complete radiologic response overall. This may limit the broader applicability of these findings, and we cannot make any definitive claims about how other patients might respond to the aforementioned treatment regimens. There was a range of LRTs used in the study which may have influenced patient outcomes, and patients may have responded differently to each combination of LRT + immunotherapy. The sample size precluded a statistical analysis of the factors that might have influenced either the response to this approach or recurrence-free survival after resection in this cohort. Not all patients received resection, so the degree of pathologic response could not be determined. We also recognize that ctDNA was not obtained in a consistent fashion for the included patients, and that only having data on four patients limits any conclusion of ctDNA and/or TMB as a biomarker. As shown in Figure 2, we have now established standardized protocols such that we can study ctDNA and TMB as a biomarker in more robust fashion.

Pre- and post-treatment imaging for all exceptional responders is provided (Figure 3).

## 5. Conclusions

Combined immunotherapy and locoregional therapy can downstage some patients with initially unresectable HCC, allowing for eventual curative-intent surgery. The survival benefit associated with complete response seems durable up to 3 years after achieving this response, and ctDNA measurements support disease-free status in those who ultimately undergo resection. Studies should focus on identifying ideal candidates for this approach using clinical factors and liquid biopsy.

## Figures and Tables

**Figure 1 cancers-15-05220-f001:**
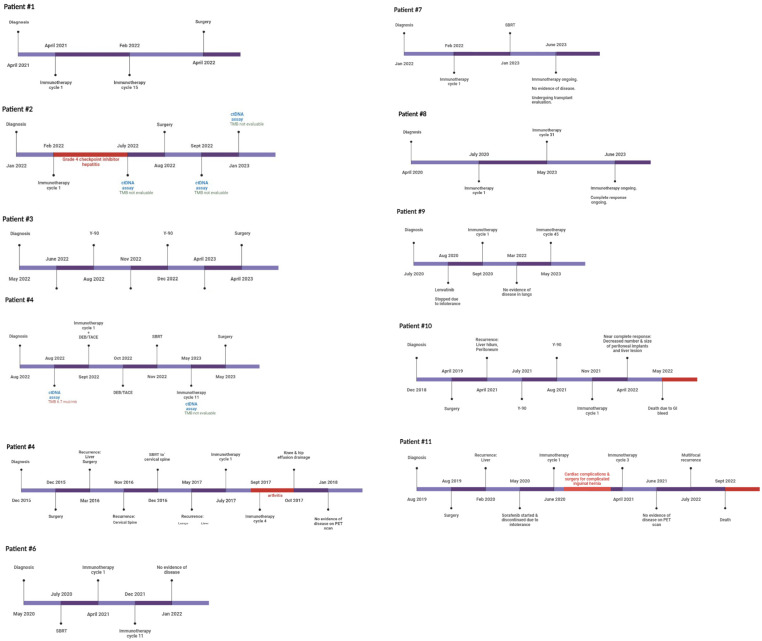
Timeline of treatment and disease response. Treatment and testing timeline for each patient with complete response to neoadjuvant immunotherapy regimens.

**Figure 2 cancers-15-05220-f002:**
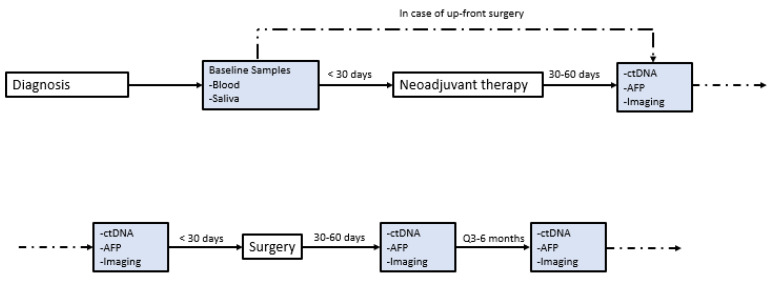
Institutional protocol for ctDNA. Institutional protocol for circulating tumor DNA (ctDNA) screening in patients with hepatocellular carcinoma (HCC) seen in our multidisciplinary liver tumor clinic. This protocol was implemented in January 2021.

**Figure 3 cancers-15-05220-f003:**
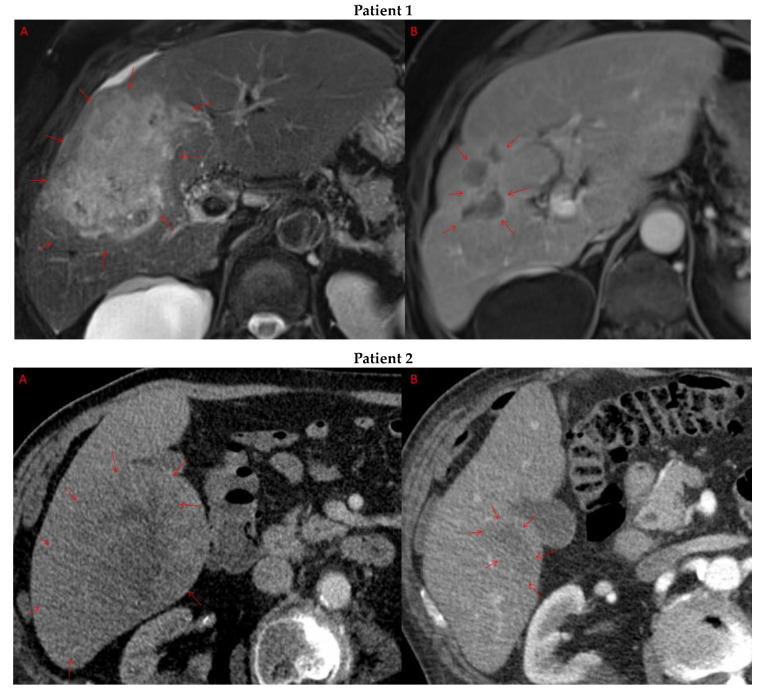
Imaging findings in patients (n = 11) with complete response to neoadjuvant immunotherapy. Cross-sectional imaging for each patient with complete response to immunotherapy. Images represent the scan most recently taken prior to beginning the course of neoadjuvant treatment and the scan taken most recently after completion. (**Patient 1**) MRI showing large pre-treatment infiltrative tumor (**A**) followed by decrease in size from 10 cm to 5.6 cm status post 15 cycles of immunotherapy (**B**). (**Patient 2**) Large ill-defined 11cm tumor at diagnosis (**A**) followed by decrease in size to 3.6cm after 1 cycle Atezolizumab (**B**). (**Patient 3**) 13.3 cm mass with periportal lymphadenopathy (**A**) and subsequent decrease in size to 6.9 cm after 5 cycles of immunotherapy+1 round Y90 (**B**). (**Patient 4**) 8.1 cm mass in segment VIII with extension to right hepatic vein (**A**,**B**), and post-treatment tumor shrinkage after 11 cycles immunotherapy and locoregional treatment (**C**). (**Patient 5**) Recurrent scattered HCC lesions after previous resection (**A**–**C**). Near complete resolution after 4 cycles of immunotherapy and 1 Y90 (**D**,**E**). (**Patient 6**) 8 cm mass in segment II/III with portal vein invasion (**A**,**B**) and complete resolution after 22 cycles immunotherapy, Y90 and SBRT (**C**). (**Patient 7**) Infiltrative 20.3 cm mass in the left lobe with extensive tumor-in-vein in portal venous system (**A**,**B**) followed by complete resolution on 20 cycles immunotherapy, locoregional treatment and SBRT (**C**). (**Patient 8**). 7.1 cm mass in the right lobe (**A**,**B**) with reduction to 1.7 cm after 31 cycles immunotherapy (**C**). (**Patient 9**) 14.7 cm mass in the right hepatic lobe (**A**,**B**) shrinking to 5.8 cm after treatment (**C**,**D**). (**Patient 10**) Patient post-partial resection of segment IV/V for HCC with recurrence demonstrating 3.2 cm segment V/VIII lesion (**A**), a 4 cm lesion in the Right lower lobe (**B**) and a peritoneal metastasis (**C**). After treatment with 9 cycles immunotherapy + multiply Y90 treatments tumor size shrunk significantly (**D**–**F**). (**Patient 11**) Patient post-right hepatectomy for HCC with multiple diffuse recurrent nodules in the liver with largest measuring 2.6 cm (**A**,**B**). Post treatment imaging showing no evidence of disease (**C**).

**Table 1 cancers-15-05220-t001:** Patient demographics.

	N = 11
Sex, N (%): Female Male	2 (18.2%) 9 (81.8%)
Race, N (%): White Black Multiracial	8 (72.7%) 2 (18.2%) 1 (9.1%)
Age at Presentation, Mean (SD)	66.7 (±9.7)
Neutrophil/Lymphocyte Ratio, Mean (SD)	3.25 (±1.20)
Total bilirubin, mg/dL, Mean (SD)	0.80 (±0.37)
Albumin, g/dL, Mean (SD)	3.65 (±0.53)
INR, Mean (SD)	1.09 (±0.08)
MELD Score, Mean (SD)	9.4 (±2.37)
Child Pugh Score, Mean (SD)	5.3 (±0.50)
Child Pugh Class, N (%) A	4 (36.4%)
Cirrhosis, N (%) Alcohol abuse history, N (%) HCV, N (%) HBV, N (%)	4 (36.4%) 3 (27.3%) 4 (36.4%) 1 (9.1%)
Prior Treatment, N (%) Ablation, N (%): Radioembolization, N (%): Chemoembolization, N (%): Radiotherapy, N (%): Resection, N (%): Systemic Chemotherapy, N (%):	7 (63.6%) 0 (0%) 4 (36.4%) 1 (9%) 3 (27.3%) 3 (27.3%) 1 (9%)
Immunotherapy, N (%): Atezolizumab + Bevacizumab Nivolumab	9 (81.8%) 2 (18.2%)
Adverse effects (immune related), N (%)	3 (27.3%)
Pre-Treatment AFP, ng/mL, Mean (SD)	28,035.2 (±58,034.2)
Pre-Treatment Number of Lesions, N (%): 1 2–3 4–10 Infiltrative Innumerable	3 (27.3%) 2 (18.2%) 3 (27.3%) 2 (18.2%) 1 (9.1%)
Pre-Treatment size of biggest lesion, cm, mean (SD)	9.25 (±5.32)
Pre-Treatment tumor vascular thrombus, N (%):	4 (36.4%)
Stage, N (%) II IIIA IIIB IVA IVB	1 (9.1%) 2 (18.2%) 4 (36.4%) 1 (9.1%) 3 (27.3%)
Post-Treatment AFP, ng/mL, Mean (SD)	32.43 (±49.39)
Patient Status, N (%): Alive Dead Cancer-Related Deaths	9 (81.8%) 2 (18.2%) 1 (9.1%)
Overall Survival (days), Mean (SD)	883 (±510)

**Table 2 cancers-15-05220-t002:** Treatment details and outcomes. M = male, F = female, OS = overall survival, NED = no evidence of disease, and pCR = pathologic complete response.

Pt. #	Gender	Age at Diagnosis	Prior/Concurrent Liver Directed Therapy	Pre-Treatment AFP (ng/mL)	Pre-Treatment Stage	Immunotherapy Agent	Number of Cycles	Immunotherapy Side Effects	Post-Treatment AFP (ng/mL)	Surgery after Downstaging	OS (Days)	Explant Pathology	Status at Last Follow Up
1	M	80	No	64,000	IIIB	Atezolizumab	15	Arthritis	106.2	Yes	748	pCR	NED
2	M	82	No	6.1	IIIA	Atezo/Bev	1	Hepatitis	16.1	Yes	469	70% necrosis	NED
3	M	51	Yes	3734.7	IVA	Atezo/Bev	5	No	158.6	Yes	341	75% necrosis	NED
4	M	73	No	72.3	IIIB	Atezo/Bev	11	No	37.1	Yes	244	75% necrosis	NED
5	M	61	No	13.8	II	Nivolumab	4	Arthritis	4.4	No	2145		NED
6	M	54	Yes	39,787	IIIB	Atezo/Bev	22	No	3	No	930		NED
7	M	67	Yes	200,000	IIIB	Atezo/Bev	20	No	4.4	No	469		NED
8	F	77	No	650	IIIA	Nivolumab	31	No	3	No	1042		NED
9	M	68	No	116	IVB	Atezo/Bev	45	No	15.6	No	979		Stable disease
10	F	63	Yes	3.9	IB	Atezo/Bev	9	No	3	No	1155		Death (Bev complication)
11	M	59	Yes	3.8	IVB	Atezo/Bev	3	No	3	No	1199		Death (disease)

**Table 3 cancers-15-05220-t003:** Summary of published case reports describing response to immunotherapy in advanced HCC.

Authors	# of Cases	Immunotherapy Agent	Response	Concurrent Local/Systemic Treatment	Adverse Effects	Surgery after Downstaging
Truong et al., 2016 [17]	1	Pembrolizumab	CR	No	No	No
Tighe et al., 2019 [18]	1	Nivolumab	PR	TACE	No	No
Chiang et al., 2019 [19]	5	Nivolumab ± Pembrolizumab	2 CR 3 PR	TACE ± SBRT	Pneumonitis dermatitis	No
Liu et al., 2019 [20]	1	Pembrolizumab	CR	Lenvatinib	No	No
Adcock et al., 2019 [21]	1	Nivolumab	CR	TARE + Sorefenib	No	No
Ando et al., 2020 [22]	1	Pembrolizumab	CR	No	No	No
Zhu et al., 2020 [23]	1	Nivolumab	CR	No	No	No
Bucalau et al., 2021 [24]	1	Nivolumab	CR	TACE	No	No
Liu et al., 2021 [25]	3	Atezo/Bev Pembrolizumab Tislelizumab	CR	No	No	No
Abdelrahim et al., 2022 [26]	1	Atezo/Bev	CR	No	No	Transplant
Goto et al., 2023 [27]	1	Atezo/Bev	PR	No	No	No
Zhong et al., 2022 [28]	1	Camrelizumab	CR	Sorafenib	No	No
Tsai et al., 2021 [29]	1	Nivolumab	CR	No	No	No
Li et al., 2021 [30]	1	Toripalimab	CR	Lenvatinib + Intratumoral cryoablation	No	No
Swed et al., 2021 [31]	1	Atezo/Bev	PR	No	Arthritis	No
Krug et al., 2022 [32]	1	Atezo/Bev	CR	No	No	No
Nong et al., 2022 [33]	2	-	CR	Lenvatinib + TACE	No	Yes
Deng et al., 2023 [34]	1	Tislelizumab	CR	RFA	No	No
Shigefuku et al., 2023 [35]	1	Atezo/Bev	CR	No	No	No

TACE: transarterial chemoembolization, SBRT: stereotactic body radiation therapy, RFA: radiofrequency ablation, Atezo: atezolizumab, and Bev: bevacizumab.

**Table 4 cancers-15-05220-t004:** ctDNA testing pre- and post-treatment course for patients (n = 4) with available testing.

Patient	Pre-Treatment ctDNA Variants (% Amplification)	Pre-Treatment TMB (Mut/mB)	Post-Treatment, Post-Resection ctDNA Variants	Post-Treatment TMB (Mut/mB)
1	CHEK2 Splice Site SNV (0.7%) ROS1 G192E (0.2%)	4.82	None Detected	0
2	TP53 L32fs (0.4%) MYC S281S (0.2%) TERT Promoter SNV (0.4%) CTNNB1 T41I (0.4%)	12.54	None Detected	0
3	MTOR T1840T (0.3%) CCND2 T280N (0.3%) ARID1A H544H (0.1%)	15.31	None Detected	0
4	PALB2 E230 (49.8%)	3.29	PALB2 E230 (49.0%)	2.73

## Data Availability

The data presented in this study are available in this article.

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
