# Peer review of "Immunotherapy Plus Locoregional Therapy Leading to Curative-Intent Hepatectomy in HCC: Proof of Concept Producing Durable Survival Benefits Detectable with Liquid Biopsy"

_cancers, 2023, doi:10.3390/cancers15215220_

Round 1

Reviewer 1 Report

Comments and Suggestions for Authors

Interesting study but with limited sample size. This study should be considered only a proof of concept and this should be specified in the title, abstract and discussion.

Figure 1 is too large...it should be shortened or deleted

The KM curve makes no sense as it was presented. I would delete it

The authors should comment on the concept of post-recurrence survival after loco-regional therapy and how these combined approach could solve this problem (cite the study PMID: 25085684)

Author Response

Thank you very much for your time. Please see the attachment. 

Reviewer 2 Report

Comments and Suggestions for Authors

Thank you for the opportunity to review your manuscript on combining Immunotherapy with loco regional therapy for HCC to allow for curative-hepatectomy.  

This is a very nice paper reporting on a cutting edge topic.  Immunotherapy is emerging as a frontline treatment for many tumor types and its role in liver cancer is just being defined.  Overall, this is a relatively large patient cohort in this field with data that supports the use of combined immunotherapy and loco regional therapy in clinical practice, and adds validation for using circulating tumor DNA as a marker of tumor burden or recurrence. 

I believe the paper is nearly publishable as is.  I recommend the following edits:

Pg 1 Ln 30: define ctDNA here, not on line 38.  Also, define TMB here, instead of pg 2 ln 64

Pg1 Ln 37: define NED

Pg 2 Ln 50; is HCC the most common liver tumor, or most common 'primary liver tumor'?

Pg 2 Ln 76; define IRB here, not on Ln 82

Pg 2 Ln 91; "coordinated at" maybe should be "coordinated by"

Pg 2 Ln 92; TMB already defined

Figure 1; Simplify and minimize by removing all the diagrams of livers, lungs, etc.

Figure 2: please clarify what the dotted line from 2nd box to 4th box represents?  If possible, add time from diagnosis to neoadjuvant therapy

Pg 7 ln 107; define OS

Pg 7 Ln 117; add"years" after mean age number

Table 1; make significant figures consistent.  E.d. if 81.8% male, then make 18.2% female, instead of 18%.  

Table 1: there are 11 patients, but only 10 with races, please include the last patient

Table 1: Units are needed for laboratory values, e.g. total Bili, alb, etc.  Also, AFP value has too many significant digits, perhaps cut off the decimal numbers here.  Pre-treatment tumor site needs units (cm)

Table 2; does death count as an adverse event, not sure, but if so, please add

Table 3: list studies in order or reverse order of publication date. Remove vertical lines in front of every other line. Remove Sr# column (not particularly needed).  Define acronyms in foot note; e.g. TARA, SBRT, TACE and RFA

Pg 10 Ln 11: Patient mistakingly capitalized

Fig 3; please make arrows and letters (A, B) larger and in white to improve clarity.  

General question: could you include how the decision was made between the use of the different immunotherapies in different cases; e.g. Atezo/Bev vs Nivolumab.

Author Response

Thank you for your time. Please see the attachment. 

Round 2

Reviewer 1 Report

Comments and Suggestions for Authors

The revised version of the paper is OK. Thank you!

Author Response

Thank you very much! We appreciate your time.